

# The association between antiphospholipid syndrome and atrial fibrillation: a single center retrospective case-control study

Zechuan Zhou and Yuansheng Liu

Department of Cardiology, Peking University People's Hospital, Beijing, China

## ABSTRACT

Antiphospholipid syndrome (APS) is a systemic autoimmune syndrome characterized by arterial or venous thrombosis, pregnancy complications and thrombocytopenia. The aim of this study is to investigate the association between APS and atrial fibrillation (AF) among patients in Peking University People's Hospital. A single center retrospective study was conducted. Cases were hospitalized patients diagnosed with AF by a cardiologist while the control group patients did not exhibit cardiac diseases. The results of the study revealed that in multivariable logistic regression, APS, anticardiolipin antibody (aCL) positivity and anti-beta-2-glycoprotein antibody (anti-$\beta$2GPI) positivity are independent risk factors of AF. APS, aCL positivity and anti-$\beta_2$GPI positivity are statistically different between AF patients and non-AF patients. Forthcoming studies are needed to clarify the potential link between APS and AF.

# INTRODUCTION

Atrial fibrillation (AF) is a common clinical arrhythmia (*Pan et al., 2023*). The mechanisms underlying atrial fibrillation are diverse, such as atrial fibrosis, inflammation response, oxidative stress response, obesity, intake of alcohol, caffeine and other substances in the diet (*Sagris et al., 2021*). Among them, the mechanisms of inflammation response and atrial fibrosis share similarities (*Sagris et al., 2021*). The mediators of inflammation can alter atrial electrophysiology and structural matrix, thereby increasing susceptibility to AF in patients (*Pan et al., 2023*). Inflammation also regulates calcium homeostasis and junction proteins, which are associated with triggers for AF and atrial conduction, contributing to structural remodeling of the atria (*James et al., 1996*).

Antiphospholipid syndrome (APS) is a kind of autoimmune disease characterized primarily by recurrent vascular thrombotic events, spontaneous abortions and thrombocytopenia, accompanied by moderate to high positivity for antiphospholipid antibodies (*Ghembaza & Saadoun, 2020*). APS displays a complex and diverse nature involving vascular pathology, which may lead to potential impact on various organs and tissues (*Ghembaza & Saadoun, 2020*). Cardiac involvement is commonly identified among the diverse manifestations, encompassing conditions like coronary artery disease,

Corresponding author
Yuansheng Liu, lyspku@126.com

cardiomyopathy ,valvular heart disease and pulmonary hypertension (*Ghembaza & Saadoun, 2020*). Antiphospholipid antibodies not only initiate thrombosis but also accelerate the process, which is related to the damage of endothelial cells and inflammatory mechanisms (*Knight & Kanthi, 2022*). *In vitro* experiments have shown that antiphospholipid antibodies can activate the expression of adhesion molecules and tissue factor in endothelial cells, leading to the inhibition of anti-inflammatory transcription factors (*Knight & Kanthi, 2022*). Similarly, thrombosis in APS is also associated with anti-phospholipid antibodies and inflammatory responses (*Knight & Kanthi, 2022*). Numerous studies have been undertaken to explore the correlation between APS and cardiovascular diseases (*Pan et al., 2023*).

In a recent retrospective cohort study involving 383 patients indicated that anti-beta-2-glycoprotein 1 antibody IgG (anti-$\beta$2GPI IgG), arterial thrombosis and categorized levels of estimated glomerular filtration rate were risk factors for valvular heart disease (*Pan et al., 2023*). Antiphospholipid antibodies and complement deposition have been found on the valves of patients with APS-related valvular disease, suggesting a significant role of anti-phospholipid antibodies in the development of valvular lesions (*Zuily et al., 2013*). Additionally, the hypercoagulable state induced by APS can also lead to the formation of valvular thrombi (*Zuily et al., 2013*). The peptides bound by antiphospholipid antibodies share the same sequences with certain bacteria and viruses, suggesting that an indirect infection may contribute to valvular disease (*Zuily et al., 2013*). Valvular heart disease increases the workload on the atria, leading to atrial cardiomyopathy, atrial enlargement and abnormal atrial conduction, which result in atrial fibrillation (*Molnár et al., 2023*). But there is hardly any research continuing to confirm that APS leads to arrhythmias directly.

It is widely acknowledged that structural heart disease significantly influences arrhythmogenicity (*Pan et al., 2023*). However, the electrophysiological influences of inflammation represent a relatively novel mechanism, which has largely been overlooked thus far (*Baek et al., 2016*; *Kim, Liu & Solomon, 2014*; *Lazzerini, Capecchi & Laghi-Pasini, 2017*; *Wang et al., 2021*). Systemic inflammation may also contribute to arrhythmogenesis in APS, independent of any development of structural heart disease. Moreover, there is a study that have discussed the relationship between APS patients and abnormal QT intervals in which patients with a history of heart disease have been excluded (*Baek et al., 2016*).

However, there is still rare research indicating the link between APS and arrhythmia, especially AF. To address this gap and further explore this underlying association, we conducted a single center retrospective case-control study utilizing database of Peking University People's Hospital to assess the association between APS and AF.

## METHOD

### Study design and population

Many factors are associated with atrial fibrillation, such as inflammation, atrial fibrosis and patient lifestyle (*Sagris et al., 2021*). In the mechanism of inflammation, various inflammatory cells and cytokines are involved, altering the electrical conduction of the atrial myocardium and thereby leading to atrial fibrillation (*Sagris et al., 2021*). This

is a single center observational retrospective case-control study conducted in Peking University People's Hospital. The subjects of study were hospitalized patients diagnosed with paroxysmal AF from Peking University People's Hospital from January 2012 to September 2022 in the electronic medical record system. We collected the hospital identity documents of all patients, selected patients through simple random sampling and finally included patients based on inclusion and exclusion criteria. The inclusion criteria for cases were as follows: (1) Patients aged 18–80 years. (2) Clinicians diagnosed paroxysmal atrial fibrillation (AF) and explicitly recorded it in the electronic medical records.(3) Patients without a history of heart disease . The exclusion criteria were as follows: (1) Patients with a history of cardiac surgery, valvular disease, and heart failure. (2) Patients with a history of renal dysfunction. (3) Patients with a history of hyperthyroidism. (4) Patients with a history of cancer. (5) Patients with rheumatic immune diseases other than APS (*Zhong et al., 2022*). We included a total of 213 patients, including 106 patients with paroxysmal AF and 107 age and sex matched controls. This study was approved by the Medical Research Ethics Committee of Peking University People's Hospital (No. 2022PHB007-001). Because the data are anonymized, the Ethics Committee of Peking University People's Hospital (No. 2022PHB007-001) waived the need for informed consent.

### Study variables
Demographic and clinical variables included gender, age, body mass index (BMI), alcohol, smoking, APS (primary or secondary) and laboratory variables (troponin I(TnI), brain natriuretic peptide(BNP), aCL, anti-$\beta$2GPI, lupus anticoagulant (LA)). All data were obtained using standard hospital procedures and then reviewed through the electronic medical record system in hospital. Professional cardiologists made diagnoses and all serum indicators were tested by hospital laboratories.

### Definition and identification of paroxysmal AF
According to guidelines, the definition of paroxysmal AF is an episode that lasts $\leq$7 days (often $\leq$2 days), generally terminates on its own, and can recur. All patients with paroxysmal AF were identified and diagnosed by specialized clinicians (*Joglar et al., 2024*).

### ACL and anti-$\beta$2GPI assays
A QUANTA LiteTM detection kit (INOVA Diagnostic Inc., San Diego, CA, USA) assay was used to detect IgG and IgM aCL (GPL and MPL, respectively) and anti-$\beta$2GPI (UA/ml), using the ELISA technique. IgG and IgM aCL and anti-$\beta$2GPI were confirmed through chemiluminescence assay, using Zenit RA Immunoanalyzer (A. Menarini Diagnostics, Florence, Italy) (*Devreese et al., 2018*).

### LA test
All of the patients' plasma samples were tested for LA, which was studied in two coagulation systems. Firstly, a dilute sensitized activated partial thromboplastin time and a dilute Russells' viper venom time were performed, followed then *via* confirm test. Reagents and instrumentation were provided by Hemoliance Instrumentation Laboratory (Lexington, MA, USA) (*Devreese et al., 2018*).

**Table 1  APS diagnostic criteria.**

**Clinical criteria**

| | |
|---|---|
| Vascular thrombosis | One or more clinical episodes of arterial, venous, or small blood vessel thrombosis in any tissue or organ. |
| Pregnancy morbidity | (1)One or more unexplained deaths of a morphologically normal foetus at or beyond the 10th week of gestation; (2) One or more premature births of one or more premature births of a morphologically normal neonate before the 34th week of gestation because of eclampsia or severe pre-eclampsia; or recognized features of placental insufficiency. (3)Three or more unexplained consecutive spontaneous abortions before the 10th week of gestation, after excluding maternal anatomical or hormonal abnormalities, as well as paternal and maternal chromosomal causes |

**Laboratory criteria**

| | |
|---|---|
| Lupus anticoagulant(LA) | LA present in blood circulation, on two or more occasions at least 12 weeks apart, detected according to the guideline of the International Society on Thrombosis and Hemostasis. |
| Anticardiolipin antibody(aCL) | aCL of IgG and/or IgM isotype in serum or plasma, present in medium or high titer(>40 GPL or MPL units, or >99th percentile), on two or more occasions at least 12 weeks apart, measured by standardized ELISA. |
| Anti-beta-2-glycoprotein antibody (anti-$\beta$2GPI) | Anti-$\beta_2$GPI of IgG and/or IgM isotype in serum or plasma in titer >99th percentile, present on two or more occasions at least 12 weeks apart, measured by standardized ELISA. |

## Criteria of APS

At least one clinical criteria and one laboratory criteria must be met to diagnose APS (*Devreese et al., 2018*) (Table 1).

## Statistical analysis

The data for this study were entered by personnel who were unaware of the study's objectives. The steps of data organization were completed by the researchers. Statistical analysis was conducted using IBM SPSS version 26.0. Continuous variables that followed a normal distribution (age, BMI, TnI and BNP) were represented by mean ± standard deviation and comparisons of continuous variables were conducted using independent sample T-tests; categorical variables were expressed as frequency, percentage and 95% confidence interval (CI) and chi-square test was used to compare differences between categorical variables. Finally, univariate and multivariate logistic regression analyses were used to examine the relationships between APS, aCL, anti-$\beta$2GPI and paroxysmal AF and calculate the odds ratios (OR) of various risk factors. In univariate analysis, variables with a *p*-value less than 0.05 were included in multivariate logistic regression. And in multivariate logistic regression, stepwise method (sls = 0.1 and sle = 0.05) was used to identify independent risk factors. *P*-value <0.05 was considered statistically significant (*Nasser et al., 2015*).

**Table 2  Baseline characteristics of controls and cases.** Age, BMI, TnI and BNP were continuous variables which were represented by mean ± standard deviation and comparisons of them were conducted using independent sample T-tests.

| Characteristics | All | Controls | Cases | *P* value |
|---|---|---|---|---|
| Age (y) | 55.83 ± 15.95 | 54.76 ± 17.53 | 56.91 ± 14.18 | 0.3268 |
| BMI(kg/m$^2$) | 23.96 ± 3.02 | 24.10 ± 3.04 | 23.81 ± 3.02 | 0.4906 |
| TnI(pg/ml) | 5.46 ± 2.59 | 5.28 ± 2.75 | 5.64 ± 2.42 | 0.3042 |
| BNP(pg/ml) | 47.74 ± 24.14 | 45.78 ± 23.96 | 49.73 ± 24.28 | 0.2333 |

## RESULTS

### Description of the cohort

A total of 213 individuals were enrolled in the study: 107 (50.23%) controls and 106 (49.77%) cases. The baseline characteristics showed no significant differences between the controls and the cases (Tables 2 and 3).

There was one patient diagnosed as APS in controls, 16 patients diagnosed as APS in cases. Including patients who were not diagnosed with APS but had positive antibodies, there were two patients with aCL positivity and three patients with anti-$\beta_2$GPI positivity in controls, 25 patients with aCL positivity and 23 patients with anti-$\beta_2$GPI positivity in cases (Table 4).

APS ($p = 0.0001$), aCL positivity ($p < .0001$) and anti-$\beta_2$GPI positivity ($p < .0001$) were statistically different between patients with and without AF. They were significantly higher in cases (Table 4).

In univariate logistic regression, APS with OR = 18.83 (95% CI [2.45–144.72]; $p = 0.0048$), aCL positivity with OR =16.20(95% CI [3.73–70.40]; $p = 0.0002$) and anti-$\beta_2$GPI positivity with OR =9.61 (95% CI [2.79–33.10]; $p = 0.0003$) were risk factors of AF (Table 5).

In multivariable logistic regression, after adjusting for factors such as age, gender, smoking history, alcohol consumption history ,BMI ,TnI and BNP, APS with OR =8.86 (95% CI [1.05–74.92]; $p = 0.0453$), aCL positivity with OR =14.95 (95% CI [3.36–66.50]; $p = 0.0004$) and anti-$\beta_2$GPI positivity with OR =6.79 (95% CI 1.85–24.88; $p = 0.0038$) were independent risk factors of AF (Table 5).

## DISCUSSION

This is the first case-control study conducted to evaluate the association between APS and AF among adults in Peking University People's Hospital. From this study, we can conclude that APS, aCL positivity and anti-$\beta_2$GPI positivity are significantly higher in cases and they are independent risk factors of AF.

The mechanisms by which APS induces inflammatory responses are very complex. Anti-$\beta$2GPI can trigger platelet activation and the interaction between platelets and leukocytes can lead to the activation of immune inflammatory responses (*Knight & Kanthi, 2022*). Various monoclonal antiphospholipid antibodies can induce neutrophil activation, resulting in phagocytosis (*Knight & Kanthi, 2022*). *In vitro* experiments have

**Table 3** **Baseline characteristics of controls and cases.** Categorical variables were expressed as frequency, percentage and chi-square test was used to compare differences between controls and cases.

| Characteristics | | All | Controls | Cases | *P*value | Chi-Squared Test |
|---|---|---|---|---|---|---|
| Age (y) | <60 | 109(51.17) | 55(51.40) | 54(50.94) | 0.9466 | 0.00 |
| | ≥60 | 104(48.83) | 52(48.60) | 52(49.06) | | |
| Gender | Female | 146(68.54) | 74(69.16) | 72(67.92) | 0.8462 | 0.04 |
| | Male | 67(31.46) | 33(30.84) | 34(32.08) | | |
| Alcohol | No | 168(78.87) | 85(79.44) | 83(78.30) | 0.8389 | 0.04 |
| | Yes | 45(21.13) | 22(20.56) | 23(21.70) | | |
| Smoking | No | 148(69.48) | 80(74.77) | 68(64.15) | 0.0925 | 2.83 |
| | Yes | 65(30.52) | 27(25.23) | 38(35.85) | | |
| BMI | <28 | 202(94.84) | 100(93.46) | 102(96.23) | 0.3613 | 0.83 |
| | ≥28 | 11(5.16) | 7(6.54) | 4(3.77) | | |
| TnI | <10.44 | 209(98.12) | 105(98.13) | 104(98.11) | 0.9924 | 0.00 |
| | ≥10.4 | 4(1.88) | 2(1.87) | 2(1.89) | | |
| BNP | <100 | 209(98.12) | 105(98.13) | 104(98.11) | 0.9924 | 0.00 |
| | ≥100 | 4(1.88) | 2(1.87) | 2(1.89) | | |

**Table 4** **Association between APS and AF.** APS and antiphospholipid antibodies were represented as categorical variables and chi-squared test was used to compare the differences between the two groups.

| Variable | | All | Controls | Cases | *P*value | Chi-Squared Test |
|---|---|---|---|---|---|---|
| APS | No | 196(92.02) | 106(99.07) | 90(84.91) | 0.0001 | 14.54 |
| | Yes | 17(7.98) | 1(0.93) | 16(15.09) | | |
| aCL | Negative | 186(87.32) | 105(98.13) | 81(76.42) | <.0001 | 22.69 |
| | Positive | 27(12.68) | 2(1.87) | 25(23.58) | | |
| anti-$\beta_2$GPI | Negative | 187(87.79) | 104(97.20) | 83(78.30) | <.0001 | 17.74 |
| | Positive | 26(12.21) | 3(2.80) | 23(21.70) | | |
| LA | Negative | 200(93.90) | 103(96.26) | 97(91.51) | 0.1475 | 2.10 |
| | Positive | 13(6.10) | 4(3.74) | 9(8.49) | | |

**Table 5** **Univariate logistic regression and multivariable logistic regression.** Univariate and multivariate logistic regression analyses were used to examine the relationships between APS and AF.

| Variable | Univariate OR | *P*value | Multivariate OR | *P*value |
|---|---|---|---|---|
| APS | 18.83(2.45–144.72) | 0.0048 | 8.86(1.05–74.92) | 0.0453 |
| aCL | 16.20(3.73–70.40) | 0.0002 | 14.95(3.36–66.50) | 0.0004 |
| anti-$\beta_2$GPI | 9.61(2.79–33.10) | 0.0003 | 6.79(1.85–24.88) | 0.0038 |
| BMI | 0.56(0.16–1.97) | 0.3671 | | |
| Gender | 1.06(0.59–1.89) | 0.8462 | | |
| Alcohol | 1.07(0.55–2.07) | 0.8389 | | |
| Smoking | 1.66(0.92–2.99) | 0.0938 | | |
| TnI | 1.01(0.14–7.30) | 0.9924 | | |
| BNP | 1.01(0.14–7.30) | 0.9924 | | |

shown that antiphospholipid antibodies can induce monocytes to express tissue factors and pro-inflammatory factors, thereby promoting inflammatory responses (*Knight & Kanthi, 2022*).

The effect of aCL and anti-$\beta_2$GPI is evident in myocardial ischemia, myocarditis and valvular disease, which is related to the types and titers of antibodies (*Djokovic et al., 2022*). According to existing literature, the mechanism is not only associated with thrombosis but also includes inflammation, complement pathways, platelet activation, etc (*Kelchtermans, Chayouâ & Laat, 2018*; *Pengo et al., 2010*; *Stojanovich et al., 2013*). As previously mentioned, the deposition of anti-$\beta$2GPI and complement in heart valves can cause inflammation, immune responses (*Zuily et al., 2013*). Bacteria or viruses with the same sequence as certain peptides can bind to anti-$\beta$2GPI and cause heart damage (*Zuily et al., 2013*). Anti-$\beta$2GPI may further contribute to the occurrence of atrial fibrillation by altering the structure of heart valves. We speculate that aCL may also have a similar mechanism and different titers may have varying effects, necessitating further stratified research. However, more basic research is needed to explore this mechanism.

Many studies have shown that in the inflammatory and fibrotic mechanisms of AF, various inflammatory cells can promote the fibrosis process, such as macrophages, mast cells and Th2 cells, which are associated with the action of various cytokines such as TNF and IL-1 (*Sagris et al., 2021*). Additionally, substances involved in acute inflammatory responses, such as CRP and IL-6 also play important roles in the occurrence of atrial fibrillation (*Sagris et al., 2021*). Therefore, there may be some overlapping targets between the inflammatory response of APS and the inflammatory mechanisms of AF.

According to previous studies, the inflammatory response and irregular activation of the immune system could disrupt the electrophysiological processes in the atria, resulting in heightened excitability of atrial myocardium and increasing the likelihood of atrial fibrillation (*Baek et al., 2016*; *Bandyopadhyay et al., 2021*; *Kim, Liu & Solomon, 2014*; *Lazzerini, Capecchi & Laghi-Pasini, 2017*; *Wang et al., 2021*). These changes may involve modifications in the action potentials of atrial myocardial cells and influence the atrial conduction system (*George et al., 2007*). The presence of anti-phospholipid antibodies may lead to inflammation and damage to myocardial cells, subsequently affecting the structure and function of the myocardium (*Tincani et al., 2006*). These changes may increase susceptibility to atrial fibrillation, making atrial myocardial tissue more prone to irregular excitability (*Tincani et al., 2006*).

Additionally, it has been found in previous studies that LA is a major risk factor for thrombosis events (*Tincani et al., 2006*), but there is no definitive evidence to suggest that they can directly damage the heart.

To improve the quality of future research, it is crucial to establish larger multicenter cohorts and utilize prospective studies that incorporate a broader range of indicators for observation. In this study, many atrial fibrillation patients lacked results from antiphospholipid antibodies and other autoantibody tests, inevitably introducing bias. In future research, in addition to comprehensive serum marker testing, we could incorporate cardiac imaging examinations to more comprehensively observe patient outcomes. After establishing a multicenter prospective cohort, predictive models can be developed to guide

the prediction and treatment of AF in APS. More basic research is also needed to explore the specific mechanisms between the two diseases.

In conclusion, the cardiac manifestations of APS are diverse and may be associated with certain types and levels of antiphospholipid antibodies. This need further prospective research to confirm these associations. Our study provides insights for future research, serving as a starting point to enhance attention to cardiovascular complications in autoimmune diseases.

# CONCLUSION

In conclusion, our study indicated that APS, aCL positivity and anti-$\beta_2$GPI positivity were significantly higher in cases and they were risk factors of AF. These findings imply that types and levels of antiphospholipid antibodies may contribute to this result. Therefore, it would be necessary to design a prospective cohort study to further explore the aforementioned relationship.

## Funding
The authors received no funding for this work.

## Competing Interests
The authors declare there are no competing interests.

## Author Contributions
- Zechuan Zhou conceived and designed the experiments, performed the experiments, analyzed the data, prepared figures and/or tables, authored or reviewed drafts of the article, and approved the final draft.
- Yuansheng Liu conceived and designed the experiments, prepared figures and/or tables, and approved the final draft.

## Human Ethics
The following information was supplied relating to ethical approvals (*i.e.*, approving body and any reference numbers):

The Peking University People's Hospital granted Ethical approval to carry out the study within its facilities (Ethical Application Ref: 2022PHB007-001).

## Data Availability
The raw measurements are available in the Files S1 and S2. The raw data shows the information of the controls and cases

## Supplemental Information
Supplemental information for this article can be found online at http://dx.doi.org/10.7717/peerj.17617#supplemental-information.

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
