# Peer review of "The association between antiphospholipid syndrome and atrial fibrillation: a single center retrospective case-control study"

_PeerJ, doi:10.7717/peerj.17617_

## Round 0.1 · original submission · Major Revisions

The abstract mentions that the study aims to investigate the association between APS and atrial fibrillation (AF). However, the title suggests that the manuscript will discuss how antiphospholipid syndrome affects the prognosis of arrhythmia. Please ensure that the title accurately reflects the study's aim.

1. Please improve the description of antiphospholipid syndrome;
2. Please clearly state the inclusion and exclusion criteria, and it's better to use flowchart;
3. Please provide more detail on the definition and identification of paroxysmal AF and the criteria of APS to ensure reproducibility.
4. Please clarify the criterion for variable inclusion in the multivariate logistic regression.
5. I think it better to convert some text into tables and include a figure summarizing the role of APS in AF patients.

Reviewer 1 ·

Basic reporting

The authors present article to discuss the association between APS and atrial fibrillation. I look forward to reviewing the updated and improved manuscript. Comments and suggestions are as follows:
1. Modify the title to increase reader interest. Consider less absolute wording.
2. In the abstract, change the expression of confidence intervals to "to" for better understanding.
3. Introduction:
• Provide a reference to support the statement in line 48.
• Elaborate on the inflammatory mechanisms of anti-phospholipid antibodies.
• Explain in detail the mechanisms by which APS leads to thrombosis and subsequent cardiovascular complications.

Experimental design

4. Methods:
• Clearly state the inclusion and exclusion criteria.
• Describe the APS patients in detail, including the number of patients diagnosed with APS and those with only positive antibodies.
• Specify the antibody detection methods and diagnostic criteria for APS.
• Describe the mechanisms of atrial fibrillation occurrence from various perspectives.
• Further elaborate on the inflammatory mechanisms underlying atrial fibrillation.
• Explain the mechanisms by which APS-induced valve disease leads to arrhythmias.

Validity of the findings

5. Results:
• Provide detailed information about the 213 included patients in the supplementary material.
• If more baseline indicators were analyzed, present them in the results section.
6. The discussion section needs to be revised and improved based on the methodological enhancements.
7. Carefully proofread the entire manuscript for grammar and spelling errors, such as "anti-β2GPI" instead of "anti-β2-GPI."
8. Consider converting some text into tables. Include a figure summarizing the role of APS in AF patients.
9. Describe the data analysis methods in more detail, such as reporting the assessment of the inconsistency assumption using the unrelated mean effects (UME) model for global inconsistency and node splitting for local inconsistency.
10. In the discussion section, explore whether specific types and concentrations of anti-phospholipid antibodies can predict the occurrence of atrial fibrillation.
11. If more variables can be included, conduct analyses from multiple perspectives to elucidate the relationship between APS and atrial fibrillation.

Reviewer 2 ·

Basic reporting

The theme of the manuscript is very relevant, and the language is generally appropriate. However, I have some suggestions to enhance the basic reporting. There is a spelling mistake in the title of Table 1 and in line 111, "ctiteria."

The title suggests that the manuscript will discuss how antiphospholipid syndrome affects the prognosis of arrhythmia. However, it is not the aim of the manuscript. The purpose of this study is to investigate the potential association between antiphospholipid syndrome and the occurrence of atrial fibrillation.

In the abstract, the sentence on lines 41 and 42, "More treatment measures must be found to reduce the incidence of atrial fibrillation in APS." does not seem to fit because it is not discussed in any part of the manuscript. I suggest excluding this sentence.

I believe that, in the Abstract, the use of abbreviations should be avoided, such as aCL and anti-B2GPI IgG. I believe that in the abstract, the use of abbreviations should be avoided, such as aCL and anti-B2GPI IgG.

Your introduction requires more detail. I suggest improving the description of antiphospholipid syndrome in lines 50 and 51 to enhance the background of your study.

Also, in the introduction, the abbreviation anti-B2GPI IgG is not explained. Usually, when an abbreviation is used for the first time, it should be explained for further understanding. Review the method as well (BMI, line 91).

In the introduction, lines 68-69, the author states, "(...) in many studies that have confirmed the relationship between APS patients and abnormal QT (...)," although only one reference is cited. Please add additional references or revise the sentence.

Some information is missing from the tables. A legend should be included, providing explanations for the abbreviations used and detailing the statistical test employed. The title of Table 1 is very vague. In tables 3 and 4, the column title " statistics" should be replaced with the measure used. I did not understand the results presented in that column.

Add a capital letter on line 85 in the sentence ".We included total 213 ...".

On line 130 of the results section, the sentence "Characteristic of the Study Sample." does not fit the context. Please review. On lines 132- 133, the results section, the sentence "There was 1 patient diagnosed as (...)" is confusing. Please rephrase.

Around 23% of the references are more than 10 years old (2,12,13 and 17), and reference number 2 from 1996 is used to provide a definition of atrial fibrillation. When applicable, I suggest using more recent references.

Experimental design

The authors provided ethical approval in a language other than English. I suggest translating it into English.

In the Methods section, "Definition and identification of paroxysmal AF" and "Ctiteria of APS" should be better developed with more detail to ensure the reproducibility of the manuscript.

In statistical analysis, what was the criterion for variable inclusion in the multivariate logistic regression? In univariate analysis, variables with a p-value less than or equal to 0.2 are typically included in multivariate logistic regression.

Validity of the findings

The results of OR in the manuscript should be reported with the p value (lines 37-8; 142-148). I suggest the following format: OR= 8.86 (95% IC 1.05- 74.92; p= 0.0453).

Additional comments

The manuscript has a significant scope, but some changes are necessary to enhance its clarity and relevance.

---

## Round 0.2 · Minor Revisions

The manuscript has undergone revisions and shows varying degrees of improvement across all sections. However, it still has quite a few issues and does not yet meet the requirements for publishing a high-quality paper. The authors are advised to further revise and refine the title, abstract, introduction, methods, results, and discussion sections of the article based on the reviewers' comments.

Reviewer 1 ·

Basic reporting

The authors have made efforts to address the comments and suggestions provided in the previous review. However, there are still some areas that require further improvement:
1. The title has been modified, but the current phrasing "The Discussion of Relationship between..." is still awkward. Consider further refining the title for clarity and conciseness.
2. In the abstract, the expression of confidence intervals has been corrected. However, the conclusion section of the abstract remains verbose and lacks clarity in its logical flow.
3. The introduction has been expanded with additional background information on antiphospholipid syndrome (APS) and some new references. However, the mechanisms by which APS leads to valvular heart disease and subsequently causes arrhythmias are not clearly explained. The roles of 5-HT and DA in the pathogenesis of APS also need to be discussed in more depth.

Experimental design

1. The methods section now includes more information on the inclusion and exclusion criteria and provides some details on the diagnostic criteria and detection methods for APS, aCL, and β2-GPI. However, the description of the mechanisms of atrial fibrillation occurrence and the inflammatory mechanisms underlying atrial fibrillation remains insufficient.
2. The results section has been updated with a more detailed description of the general characteristics of the 213 study subjects. However, it seems that no additional baseline indicators have been included in the analysis. The modifications to the tables are still not standardized.

Validity of the findings

1. The discussion section has been expanded but fails to provide an in-depth discussion based on the methodological improvements. The relationship between different antibody types and titers in APS and atrial fibrillation is not thoroughly explored. The analysis of the relationship between APS and atrial fibrillation lacks a multi-faceted and multi-level perspective.
2. The references have not been sufficiently updated, with some important references still being relatively old.

Additional comments

The variable screening criteria for multivariate logistic regression analysis have been added to the statistical methods. However, details of other statistical analyses are not provided.

Reviewer 2 ·

Basic reporting

No comment

Experimental design

No comment

Validity of the findings

No comment

Additional comments

No comment

---

## Round 0.3 · accepted · Accept

The authors have addressed all comments and made revisions again. I have no any other questions.